# Experimental study of embankment breach based on its construction parameters

Sachin Dhiman<sup>1</sup>, Kanhu Charan Patra<sup>1</sup>,

<sup>1</sup>Civil Engineering Department, National Institute of Technology Rourkela, Rourkela, 769008, India

Correspondence to: Sachin Dhiman (sach.dhiman1988@gmail.com; 514ce1012@nitrkl.ac.in)

**Abstract.** Laboratory data obtained from the overtopping failure of eight cohesive embankments built with different construction parameters are presented in this paper. Experiments were performed in two phases. Five experiments under phase 1 were carried out in small width flume having dimension viz. 17 m long, 0.6 m wide and 0.6 m high. In phase 2, three experiments were performed on large width flume having dimensions viz. 13 m long, 1.75 m wide and 0.5 m high. The

- construction parameters which were varied in embankments are compaction effort, compaction moisture content, and the moisture content at the time of failure. Phase 1 investigate the effect of compaction effort and compaction moisture content on breach parameters. As the water content of clay decreases or increases, the soil shrinks or swells which can cause damage ranging from small hairline cracks to severe structural distress. The sudden rise of water level in the reservoir due to heavy rain will cause rushing of water through cracks and if water overtopped the embankment will lead to quick failure. An
- attempt was made in phase 2 to investigate the effects on breach evolution when the embankments were air dried to reduce its moisture content and to develop cracks. The final breach shape and parameters noted in experiments show remarkable change for each dam failure. In the remainder of this paper, we report results of a set of small and large scale damovertopping experiments that were designed to; (1) investigate the effect of changing compaction effort and moisture content of the headcut migration, breach parameters, and flood hydrograph; (2) assess the overtopping behaviour of embankments
- when they are air dried for a long time.

#### **1** Introduction

For every structure, there is a limit to which it can build to withstand the forces applied. An unexpected and unforeseen natural event increases the applied forces that might trigger the failure of the structure. Potential dam failure is one of such events which produce flash flood that causes significant loss to human lives and property. In early 1960's there was a

25 growing awareness worldwide of the devastating consequences due to a potential dam failure. Most of the failures noticed in the records were breaching of earthen embankments due to overtopping. Therefore significant efforts have been made in detail the complex mechanism and phenomena of embankment breaching, but still many uncertainties related to breach modeling are still existent (Morris 2005). Incomplete understanding of breach formation processes leads to the limited capabilities of the mathematical description of breaching mechanisms thus presently developed breach models rely on

several assumptions. One of the vital steps in dam break modelling is the accurate prediction of the breach outflow hydrograph and towards this many fields and laboratory experiments were carried out for different aspects like embankment geometry, construction parameters, soil texture and hydraulic conditions. The prediction of geometric and temporal parameters of embankment dam breaches is widely recognized as one of the most significant downstream flooding and associated consequences (Wahl et al. 2009).

The rate of breach development has a significant impact on the peak discharge from a dam failure (Hanson et al. 2008). Based on the 18 historical cases of earthen embankment dam failure, Walder O'Connor (1997) provided some insight into the rate of breach development. They observed the mean vertical erosion rate determined from the historical cases ranged from 1 to 1000 m/h. White and Gayed (1943) conducted laboratory overtopping tests on 0.3 m high embankments and

- observed that erosion rate of cohesive soil embankments varied too much that it is not possible to be correlated numerically. However, they noted that the erosion rates are susceptible for clay and water percentages. Powledge and Dodge (1985) noticed that the erosion of small embankments in flume tests reduced by half with the increase in compaction from 95% to 102% of Standard Proctor compaction. Hassan et al. (2004) made out that increase in compaction effort and compaction water content for the same soil increased the erosion resistance of small embankments significantly. Hunt et al. (2005),
- performed three large-scale tests to investigate breach widening rate. Two embankments were constructed using silty sand having same dry unit weight, but with different moisture content and the third experiment was made of lean clay. They concluded that breach widening rate for silty sand increased by 3.3 times faster for the 12% decrease in compaction moisture content from optimum conditions. It was expected that for lean clay test the erosion rate was much slower due to the high cohesive forces. Hanson et al. (2005) researched the overtopping failure of eight cohesive embankments. They suggested
- that breach migration rates and erosion widening rates show a direct correlation to the compaction water content, dry unit weight, and soil texture. They demonstrated that a small change in compaction water content could increase the breach widening rate by almost two orders of magnitudes.

The above studies demonstrate the importance of compaction effort and compaction moisture content on the evolution of breach parameters and breach flood. There are series of efforts going on to improve breach simulation technologies, by

25 focusing on applying physically- based models of embankments by their soil properties and construction parameters (i.e., compaction effort and compaction moisture content). It has been noticed that development of small inline cracks in earthen embankments is common. This is due to the inability to control the moisture content while constructing earthen embankments. If the embankments are built beyond the shrinkage

to control the moisture content while constructing earthen embankments. If the embankments are built beyond the shrinkage limit of soil, the possibility of crack formation increases. Further sudden drawdown and rise in water level at summer and rainy season respectively is common in reservoirs, which causes the expansion, contraction, and sliding of the soil of

30 rainy season respectively is common in reservoirs, which causes the expansion, contraction, and sliding of the soil of embankments. The exposed portion of the embankment to the direct sunlight for extended duration remove the moisture content to a greater extent that it produces small cracks on the surface of the upstream and downstream slope. The sudden rise of water level in the reservoir due to heavy rainfall cause rushing of water through cracks which weaker the bond between soil particles. If water overtopped, this may lead to quick erosion of embankment soil. The present work, through a

series of 8 types of experimental dam failure study focuses on; (1) investigation of the effect of changing compaction effort and moisture content of the headcut migration, breach parameters, and flood hydrograph; (2) assessing the overtopping behaviour of embankments when they are air dried for a long time so that they develop cracks in there surfaces.

# 2 Experimental setup and procedure

- Eight experiments of overtopping failure of cohesive embankments were carried out in two phases. Five experiments were conducted in phase 1 in a small width flume having a dimension of 17 m long, 0.6 m wide and 0.6 m high. In phase 2, three experiments were carried out on large width flume having aspects of 13 m long, 1.75 m wide and 0.5 m high at the Hydraulics Engineering Lab of NIT Rourkela. Figure 1 shows the schematic diagram of experimental setup for Phase 1 and Phase 2. A recirculating water supply system was established from an underground sump to an RCC overhead tank into the
- flumes and back to the sump through the volumetric tank. The operating valves were calibrated for different openings to get the desired discharge before the experiments. In all the experiments of phase 1, a constant inflow of 0.9 L/s was maintained, while in Phase 2 experiments inflow hydrograph was provided to the reservoir for maintaining constant head condition throughout the experiment. For obtain continuous video documentation of each experiment, two video cameras were used; one camera was placed in front of downstream slope of the embankment to provide an oblique view, while another was set at
- the top of embankment crest to provide the top view of dam crest.

#### **3** Embankment soil properties

Two types of lean clay – CL cohesive soil were used in the construction of embankments. Lab tests were performed for the soil samples to know the properties before utilizing it for the construction of the embankment. The grain size distributions of the soils were obtained from sieve analysis and hydrometer analysis [IS: 2720 (Part 4) – 1985 (Reaffirmed-2006)]. The grain

- size distribution of the soil used in phase 1 embankment construction had 29% clay, 57% silt, and 14 % sand. In case of phase 2 embankments, the particle size distribution was 26% clay, 48% silt and 26% sand. The average particle size (*d<sub>50</sub>*) of soil type 1 and soil type 2 was 0.018 mm and 0.019 mm respectively. The gradation curves for both soils are shown in Fig. 2. Before the experiments the standard proctor test [IS: 2720 (Part 8) 1983] was performed for constant compaction effort on soils to obtain the maximum dry density and optimum moisture content (OMC). These soil properties were necessary to
- estimate the amount of soil required for embankment construction to achieve the desired amount of compaction. More lab tests were performed on soil samples as per the Bureau of Indian standards to find out the plasticity index (*Ip*), specific gravity (*Gs*), cohesion and friction angle ( $\Phi$ ). The details of soils properties are given in Table 1.

#### 4 Embankment construction

In all eight experiments, the embankment height, h of 0.3 m and crest width 0.1 m were kept constant. In phase 1 30 experiments, the embankment length was 0.6 m with upstream slope of 2.5:1 (Horizontal: Vertical) and downstream slope of 2H : 1V were maintained while for phase 2 experiments, the embankment length of 1.72 m, upstream slope of 3H : 1V and

downstream slope of 2.5H: 1V were fixed. The storage capacity of the reservoir at crest level was 0.9075 m<sup>3</sup> for phase 1 and 3.4779 m<sup>3</sup> for phase 2 embankments. A rectangular notch cut of 0.03 m deep and 0.1 m wide was created at the centre of embankment crest before the start of each experiment. The details of geometry for phase 1 and phase 2 embankments are incorporated in Table 2.

- The embankments were built in six horizontal layers of 0.05 m each, compacted with manual soil temper of 30 x 10 (length x width). Before compaction, the required amount of water was mixed thoroughly with dry soil to attain the moisture content and emphasis was given for maintaining constant drop height throughout the dam construction for achieving uniformity. After achieving the predetermined height of the dam, it was trimmed to get the desired shape on the top width, upstream and downstream slope of the embankments. Fig. 3 shows few images of embankment construction for Phase 1.
- Phase 2 embankments were compacted for known moisture content and after the completion of the embankment; they left for air dried for few days to remove their moisture content to nearly half of the compacted water content. Table 3 gives the detailed construction parameters of test embankments for Phase 1 and 2 along with the relative compaction and relative water content achieved during the construction. Relative compaction is the ratio of dam dry density and proctor test dry density. Similarly, the relative moisture content is the ratio of the water content of embankment at the time of failure to the
- optimum moisture content of the soil.

### 5 Experimental results and observations

#### 5.1 Breach Evolution Stages

It was observed from the literature that the breach evolution patterns of non-cohesive test embankments are entirely different from the cohesive test embankments. Similarly, for cohesive embankments constructed with different parameters, the breach

- evolution is different. The process of breach evolution observed in Phase 1 and Phase 2 experiments for overtopping failure of cohesive dams have been fortifying into two crucial scenarios. In first scenario, the reservoir storage water was not contributing to the breached outflow, but in second scenario, it adds to the outflow from the breached portion of the embankment. The two breach scenarios can be explained in three breach stages.
  - Breach stage I: Initially when the flow starts overtopping the dam, it results in a sheet and rill erosion forming stepped or
- cascading over falls leading to headcut erosion. Headcut erosion continues until it reaches from downstream slope to the upstream crest. Duration from the beginning till the end of breach stage I is known as breach initiation time. Wahl (1988) defines the breach initiation time as the first flow over or through a dam that will initiate a warning, evacuation, or heightened awareness of the potential for dam failure.
  - Breach stage II: During this stage lowering of dam crest and erosion of upstream slope starts and continues until it reaches
- the dam height.
  - Breach stage III: In this stage widening of the breach as well as erosion of upstream slope occurs.

Time duration encompasses of breach stage II and III is known as breach formation time. At this time the crest of embankment erodes vertically with the lateral movement of breach width. Headcut also continues to advance until the soil erosion becomes insignificant, maintaining equilibrium between tail water and reservoir water level. Wahl (1988) also defines the breach formation time as the lapse between the first breaching of the upstream face of the dam until the breach is

5 fully formed. The failure of cohesive embankment P1.2, constructed with 50% reduction in moisture content for the same compaction effort behaves similarly to the non-cohesive embankments. The apparent cohesion caused by pore pressure led to near vertical breach walls during the experiment that lasted for a short period.

## 5.2 Experimental Study in Phase 1

Compaction efforts applied for the construction of embankment 1 using a 10 kg rectangular rammer and a drop height of 30 cm was 5.0 kg-cm/cm<sup>3</sup>. The dry density achieved was 22% more than the standard proctor compaction maximum dry density 10 of soil. The experiment termed as P1.1 was run for more than 24 hours at different inflow to the reservoir. It was interesting to note that in test P1.1, there was no sign of any erosion or overtopping breach throughout the experiment. The embankment P1.2, P1.3, and P1.5 were constructed for the same compaction effort of 2.9 kg-cm/cm<sup>3</sup> but with different compaction moisture content. While embankment P1.4 was built for optimum moisture content same as embankment P1.5 but with lower compaction effort of 1.4 kg-cm/cm<sup>3</sup>. The results of Phase 1 experiments are incorporated in table 4. 15

#### 5.2.1 Headcut Migration, Breach time and Breach width

The headcut migration rate for all the experiments was obtained and analysed from the recorded videos of embankment failures. Snapshot of headcut locations for experiment 2 and 3 for different time lapses are shown in Fig. 4. The cascading waterfalls last long in the embankment compacted with 75% of moisture content than the embankment with 50% moisture

- content. As seen in Fig 5 at time t = 6 min, cascaded water fall observed in the P1.3 experiment, while in case of P1.2 20 experiment a single smoothed water fall is present. It has been noted from the experiments that reduction in moisture content from the optimum moisture content by 25% and 50%, the evolution pattern was different with a very slight change in headcut migration rate.
- The dry density of P1.4 embankment was 15.4% less as compared P1.2, and it was quite impressive that the headcut 25 migration rate of P1.4 embankment was six times slower as compared to P1.2. As in dried soil, water menisci form at grain contacts, and these are under negative capillary pressure. When capillary forces are over a unit area, we refer to "matric suction stress," which creates interparticle forces. This apparent cohesion will disappear when the unsaturated soil is made saturated which leads to the fast dislodging of soil particles, i.e., increase in erosion rate. The embankment P1.2 soil was unsaturated, and there was apparent cohesion present due to the pore-water pressure. The snapshot for the P1.2 experiment
- (50% reduction in moisture content) and P1.4 experiment (50% reduction in compaction effort) at 37 min is shown in Fig. 5. 30 Headcut migration rate for P1.2, P1.3, and P1.4 are shown in Fig. 6.

For a decrease of 24% in compaction moisture content from the optimum moisture level, there was ten times increase in average headcut migration rate. Further reduction in moisture content to 50% shows little change in headcut migration rate. In first breach scenario of embankment, the headcut migration rate for experiment P1.3 was slow as compared to the headcut migration rate in experiment P1.2 but in second breach scenario, there was an increase in headcut migration rate as well as

surface erosion for P1.3 experiment as compared to experiment P1.2 was noticed leading to the quick failure of the embankment. The above discussion was observed in the variation of breach initiation time and breach formation time as shown in Fig. 7. The experiment P1.2 and P1.3 conclude that the compaction moisture content played an important role in soil erosion.

The cracks due to undercutting of breach channel side slope by water was observed in unsaturated embankments (P1.2 and

- P1.3), which leads to the overhang material (Fig 5 (a)) and when the holding capacity of overhang materials decreased it fail into the breached channel (mass erosion) leading to the partial blockage. Mostly, this blockage was at the downstream of the embankment and did not obstruct the flow as much. This mass erosion leads to the sudden increase in breach width. A similar observation was reported on non-cohesive embankments by Pikert et al. (2011). The breach width expansion rate in saturated embankments (P1.4 and P1.5) was negligible and remains constant as of the pilot channel width provided at the top
- of embankment crest.

#### 5.2.2 Breach discharge and Inflow

The breached discharge from the embankment was computed from the observed reservoir water level using water-volume balance equation (Morris et al. 2007). For the calculation of outflow hydrograph; the inflow to the reservoir and rate of change of reservoir volume was required. The inflow provided to the reservoir was constant; it means the water head keeps

- on decreasing with the progression of the breach. It can be seen that time duration of outflow hydrograph of experiment P1.2 and P1.3 is nearly same so, plotted in a single graph [Fig. 8 (a)]. The outflow hydrograph for experiment P1.4 and P1.5 are shown in Fig. 8 (b) and (c) respectively. A constant inflow of 0.9 L/s was maintained throughout the experiment P1.3 and P1.4. For experiment P1.2 and P1.5, the inflow was 0.74 l/s, and 1.4 L/s was maintained respectively. The breach outflow was strongly dependent on the construction parameters. It was observed from the overtopping failure of unsaturated
- embankments P1.2 and P1.3 that there was 2 times increase in the peak outflow from their respective inflow peaks. It has been observed that reduction in compaction water content from 25% to 50% in embankment construction, the difference in peak outflow and time to peak was very less. Maximum change in peak outflow and time to peak was noticed in embankments compacted for more than 75% moisture content. There were 20 times decrease in time to peak when 25% reduced the compaction moisture content from the optimum moisture content. Further reduction of 25% in moisture content
- during construction decreases time to peak by 7.6%. Reduction of compaction water content from 25% to 50% shows a little change in peak outflow and time to peak. More investigation on the overtopping failure of embankments constructed between the ranges of 0 to 25 % reduction in compaction water content from the optimum moisture content is required. For the saturated embankment P1.5, compacted with compaction effort of 2.9 kg-cm/cm<sup>3</sup>, there was 12.8% increase in outflow

peak noticed from the peak inflow. Similarly, for saturated embankment P1.4, compacted with 1.4 kg-cm/cm<sup>3</sup>, there was 44.4% increase in peak outflow value was observed. There were 2.6 times decrease in time to peak when the value of compaction effort was reduced to half. From Fig 8 (b) and (c), we can observe that there is a number of peaks present in the outflow hydrograph. Instead of smooth erosion in saturated cohesive embankments sometimes soil erodes as block rather then simple meticle of soil leading to much surflow.

than single particle of soil leading to sudden increase in peak outflow.

# 5.3 Experimental Study in Phase 2

In phase 2, the embankments were compacted with known water content and then air dried to form cracks in the surface. The compaction effort applied while constructing the embankment P2.1 and P2.2 was 1.13 kg-cm/cm<sup>3</sup> with the compaction moisture content of 10% and 14 % respectively. The embankments P2.1 and P2.2 were left to dry through the natural process

- for 240 hours to remove its water content for developing cracks, which were visible to naked eyes. During this duration, the embankments were made wet and dried several times. The water content of embankment P2.1 and P2.2 after air dried was nearly half of the compaction water content. The cracks developed in embankments P2.1 and P2.2 are shown in Fig 9. The construction parameters are shown in Table 3. The soil samples were collected from the top surface and inner layers of the embankment for finding out the moisture content present at the time of the experiment. The water content of embankment
- P2.1, P2.2, and P2.3 at the time of experiment was found to be 5.2%, 7.4%, and 9.8% respectively. Phase 2 experiments were performed with a constant head of water, and for maintaining constant head, an inflow hydrograph was provided into the reservoir. The breached outflow hydrograph was computed from the volumetric tank placed at the end of the flume. The sole purpose of the experiments in phase 2 was to understand the overtopping breach evolution when there were cracks present in the embankment. Phase 2 experiments results are shown in Table 5

#### 20 5.3.1 Experiment P2.1

For this test, the time taken for complete breach development was 15 min. Breach initiation time observed was 4.3 min and breach formation time was 10 min. The top and bottom breach widths formed were 76 cm and 26 cm respectively. The average breach width of the embankment was 51 cm. In the present test, the top breach width was 2.5 times of dam height. The final breach height formed was 30 cm equal to the modeled dam height. There was 15.75 % increase in the outflow peak

from the inflow peak. The time difference between the inflow peak and outflow peak is known as lag time. When breach formation takes place, the reservoir water starts contributing to the breached outflow causing the higher peak values of outflow. The attenuation and lag time observed was 3758.5 cm<sup>3</sup>/s and 120 sec respectively. The comparison between the inflow hydrograph *verses* the outflow hydrograph is shown in Fig. 10 (a).

#### 5.3.2 Experiment P2.2

The overtopping depth of flow over the embankment top was 2 cm. The total breached development time was 16 min. The Breach initiation time was observed to be 2.5 min and breach formation time was 13 min. The embankment was failed from

the corners, i.e., flume wall sides creating average breach width of 34 cm at left and 12 cm at right. The total breach width formed was 46 cm which was 1.5 times the dam height. The breach height at the right and left side of the dam were 25 cm and 22 cm respectively. In this case, the final breach height is less than the dam height by 16.66%. Percentage increase of peak outflow hydrograph from the peak inflow hydrograph was 64.98%. In this test, the attenuation of peaks observed was

5 6.9 L/s. Fig 10 (b) shows the comparison between inflow hydrograph to the outflow hydrograph. As the compaction moisture content was more than the optimum moisture content, the development of large cracks in the embankment was observed. The cracks formed were approximately 3 mm wide, 5 cm deep and the effect of these cracks on breach time, breach evolution, and peak outflow was observed.

## 5.3.3 Experiment P2.3

- The time taken for complete failure or breach formation was 35 min. Breach initiation time and breach formation time was 10 min and 24 min respectively. The final breach depth was 27 cm. The crest height of embankment after overtopping was 22 cm, i.e., up to 8 cm of crest height along the whole length of the embankment was eroded. The formation of final breach width can be explained and understood by dividing the breach process into three parts. Part 1, was the upstream slope; Part 2, was the crest width; and Part 3, was the downstream slope of the embankment. Breach width noticed in part 1 at a distance
- of 10 cm, 30 cm, and 60 cm from the upstream toe end was 48 cm, 30 cm, and 27 cm respectively. At the center (part 2), breach width was 35 cm, and in part 3, breach width at 10 cm and 40 cm from the downstream slope toe end was 54 cm and 40 cm respectively. The breach shape geometry formed in this test was shown in (Fig. 11a & 11b). For further analysis, we take an average of breach width, of 40 cm as final breach width (Table 5). The inflow hydrograph of 50-minute duration was provided to the reservoir for maintaining the constant head. There was 28.42% increase in the outflow peak was observed
- from the inflow peak. The attenuation and lag time observed was 7781.55 cm<sup>3</sup>/s and 6 min respectively. Figure 10 (c) show comparison between inflow hydrograph w.r.t the outflow hydrograph.

## 5.2.4 Analysis

The headcut migration rate can be taken of inversely proportional to the dry density of embankments. The high dry density of embankment has a slow rate of erosion which was contradictory in case of embankment P2.2. Despite the large dry

- density of P2.2, the headcut migration rate was 2.6 times faster than P2.3 (no cracks). The development of deep and long cracks in embankment P2.2 provide a path to water, that was rushed through these cracks and saturate inner dry soil completely. The apparent cohesion was reduced, as the pore pressure was reduced which in terms seen as quick failure. The rate of formation of the breach had the highest influence on the outflow discharge hydrograph. The increase in peak discharge for P2.1, P2.2, and P2.3 was 15.75%, 64.98%, and 28.42% respectively. It was observed that for the same
- compaction energy, with an increase in compaction moisture content up to optimum moisture content (dry density increases) not necessarily decreases the breach time. Crack formation plays a crucial role in the breach evolution time. There was 9% decrease in breached time for embankment P2.2 (large cracks) from the embankment P2.1 (small cracks). An increase of

16% in breach initiation time and 20% decrease in breach formation time was noticed in embankment P2.2 from the embankment P2.1.

The embankments constructed with the same compaction effort but the reduction in compaction moisture content, in general, decreases its dry density consequently quick failure of embankment. The above statement is contrary to the results of

- experiment P2.2 and P2.3. There was an increase in breach time from 15.5 to 34.35 min, a factor of 2.2 times. The reason for this behaviour was moisture content and cracks developed on the embankment at the time of the experiment. The embankments constructed with same compaction effort and moisture content but hold different experimental moisture content act different in breach evolution and time. For a decrease of 47% water content in embankment P2.1 (air dried) from the embankment P2.3, there was a decrease in breach time by a factor of 2.4 times. We can observe that moisture content of
- soil while constructing the dam has a crucial role in the formation of final breach width. A dam built with 5.3% water content, the formation of final breach width was large and breach time was less as compared to the dam made with OMC.

#### **6** Conclusions

An attempt was made to observe the breach parameters for embankments built with cohesive soil resulting from overtopping and subsequent failures. Dam failure conditions, final breach parameters, and flooding are not only dependent on reservoir

- storage, dam geometry but also reliant on the location and rate of which the breach forms. It has also been observed from the experiment that the rate of formation of breach and the final breach parameters are affected by the moisture content and compaction energy of the dam. The experimental results indicate that compaction moisture content and compaction energy of cohesive dam are important factors in deciding breach initiation time and breach formation time and more so in determining the breach width, peak outflow, and the outflow hydrograph, in line with the observation by Hassan et al.
- (2004). The breach time is more influenced by compaction energy rather than compaction moisture content, and the breach width is more dependent on compaction moisture content.

For the first time the effect of embankment cracks on overtopping breach failure was studied. It was found that the large dry density embankment erode faster as compared to low dry density embankment because of the developments of deep and long cracks in former. These cracks provide path to the water and start saturating the inner dry soil of embankment that reduces the apparent cohesion between soil particles, which in terms seen as quick failure due to overtopping.

The research observations and conclusion made in this study provide an important basis in relating the breach parameters with compaction moisture content and compaction energy for the future investigation of physical models. More studies are required to evaluate the relationship and dependence of breach parameters on compaction moisture content and compaction energy for the different geometry of dams, various types of soils and condition of dam at the time of failure.

## 30 References

Bureau of Reclamation: Downstream hazard classification guideline,. ACER Tech. Memorandum Rep.No. 11, U.S. Dept. of the Interior, Bureau of Reclamation, Denver, 1988

Nat. Hazards Earth Syst. Sci. Discuss., https://doi.org/10.5194/nhess-2017-383 Manuscript under review for journal Nat. Hazards Earth Syst. Sci. Discussion started: 13 December 2017

© Author(s) 2017. CC BY 4.0 License.

Cestero, J. F.: Experimental Investigation of Effects of soil properties on leave breach by overtopping, J. Hydraul. Eng, ASCE, HY.1943-7900.0000964, 2014.

Fread, D. L.: BREACH: An erosion model for earthen dam failures, Silver Spring, Md.: National Oceanic and Atmospheric Administration, National Weather Service, 1988.

Froehlich, D. C.: Embankment dam breach parameters revisited, Proc., Water Resources Engineering, 1995 ASCE Conf. on Water Resources Engineering, ASCE, NY, 887-891, 1995a.

Froehlich, D. C.: "Peak outflow from breached embankment dam". J. Water Resour. Plann. Manage., 121(1), 90-97, 1995a. Hanson, G. J. and Cook K. R.: Apparatus, test procedures, and analytical methods to measure soil erodibility in situ, Applied Eng. in Agric. 20(4): 455-462, 2004.

Hanson G. J., Cook K. R., and Hunt S. L.: Physical modelling of overtopping erosion and breach formation of cohesive embankments, Trans. ASAE 48(5): 1783-1794, 2005.

Powledge, G. R. and Dodge R. A.: Overtopping of small dams: An alternative for dam safety, In Proc. Specialty Conf.: Hydraulics and Hydrology in the Small Computer Age, 2: 1071-1076. Reston, Va.: ASCE, Hydraulics Division, 1985.

- Vaskinn, K. A., Lovoll, A., Hoeg, K., Morris, M., Hanson, G., and Hassan, M.: Physical modeling of breach formation: 15 Large-scale field tests, In Proc. Dam Safety 2004, Association of State Dam Safety Officials (ASDSO), CD-ROM. Lexington, Ky.: Association of State Dam Safety Officials, 2004. Visser, P. J.: Breach growth in sand-dikes, Communications on Hydraulic and Geotechnical Engineering, Report No. 98-1. Delft, The Netherlands: Delft University of Technology, Faculty of Civil Engineering and Geosciences, Hydraulic and Geotechnical Engineering Division, 1998.
- Wahl, T. L.: Prediction of embankment dam breach parameters-A literature review and needs assessment. Dam Safety Rep. No. DSO-98-004, U.S. Dept. of the Interior, Bureau of Reclamation, Denver, 1998.

Wahl, T. L.: Uncertainty of predictions of embankment dam breach parameters, J. Hydraul. Eng., 130(5), 389-397, 2004;

Wahl, T. L.: Evaluation of Erodibility-Based Embankment Dam Breach Equations, Hydraulic Laboratory Report HL-2014-02, 2014.

Walder, J. S., Iverson, R. M., Godt, J. W., Logan, M., and Solovitz S. A.: Controls on the breach geometry and flood hydrograph during overtopping of noncohesive earthen dams, AGU Publication, Water Resour. Res., 51, 6701-6724, doi:10.1002/2014WR016620, 2015.

Walder, J. S., and O'Connor, J. E.: Methods for predicting peak discharge of floods caused by failure of natural and constructed earth dams. Water Resources Res. 33(10): 1, 1997.

White, C. M., and Gayed, Y. K.: Hydraulic models of breached earthen banks, Vol. 3: The Civil Engineer in War, 181-200. London, U.K.: Institute of Civil Engineers, 1943. Xu, Y., Zhang, L. M.: Breaching Parameters for Earth and Rockfill Dams". Journal of geotechnical and geoenvironmental engineering © ASCE, 2009.