# Peer review of "Experimental study of embankment breach based on its construction parameters"

_Natural Hazards and Earth System Sciences, 2017_

## Referee Comment (RC1) · Anonymous Referee #1 · 17 Jan 2018

**GENERAL COMMENTS**

The authors present basic observations of the rates of erosion, time to breach, and observed outflows resulting from the breach of cohesive embankments constructed with different compaction conditions and post-construction moisture conditioning (drying). The Phase I studies that varied compaction effort and moisture at time of compaction do not add significantly to previously literature that has studied these same variables (e.g., several papers of Hanson, Hunt). Furthermore, the data are presented in ways that do not facilitate comparisons, since moisture contents are expressed throughout the paper as ratios to optimum conditions, not as differences from optimum, which

are typical in the field of embankment dam engineering. Efforts to relate these tests to measures for predicting erosion rates (JET erosion testing, estimating of erosion rate coefficients from compaction and moisture conditions, modeling with tools such as WinDAM or other breach models) would greatly improve the value of the paper, especially for these Phase 1 tests that are relatively similar to work that has already been done by others.

The Phase 2 studies in which cracks were allowed to develop in the dams are a new contribution to the literature, as these desiccation cracks have not been studied by others to my knowledge. While the authors give basic information about percentage changes in breach time and outflow, the actual mechanisms of erosion development in cracks and the role of cracks in accelerating the headcut and breach development process are not given much focus.

Overall, the paper is written in very fractured English and is difficult to read and understand in many passages.

SPECIFIC COMMENTS

Embankment heights are reported inconsistently throughout the paper. Values of 0.3 m, 0.5 m, and 0.6 m appear in various places. Some tests are characterized as small scale and others as large scale, despite apparently small differences in embankment height.

TECHNICAL CORRECTIONS

The works of Hanson should be more fully described. They were lab tests of constructed embankments, not studies of real dam failures. The text gives so little information that a reader could easily infer the latter.

Page 4, lines 20-23 talk of two "crucial scenarios", one in which stored reservoir water is not contributing to outflow, a second in which stored water is released through the breach. However, what follows does not describe two scenarios, but three phases

that seem to apply to all of the tests. The two "crucial scenarios" seem to never be mentioned again, suggesting that they were not so crucial. This is disturbing for the reader who feels they have missed an important point.

Units for the dimensions of the tamper equipment are not given.

———————————————

---

## Author Comment (AC1) · 26 Feb 2018

General comments:

The authors present basic observations of the rates of erosion, time to breach, and observed outflows resulting from the breach of cohesive embankments constructed with different compaction conditions and post-construction moisture conditioning (drying). The Phase I studies that varied compaction effort and moisture at time of compaction do not add significantly to previously literature that has studied these same variables (e.g., several papers of Hanson, Hunt). Furthermore, the data are presented in ways that do not facilitate comparisons, since moisture contents are expressed throughout

the paper as ratios to optimum conditions, not as differences from optimum, which are typical in the field of embankment dam engineering. Efforts to relate these tests to measures for predicting erosion rates (JET erosion testing, estimating of erosion rate coefficients from compaction and moisture conditions, modeling with tools such as Win-DAM or other breach models) would greatly improve the value of the paper, especially for these Phase 1 tests that are relatively similar to work that has already been done by others. The Phase 2 studies in which cracks were allowed to develop in the dams are a new contribution to the literature, as these desiccation cracks have not been studied by others to my knowledge. While the authors give basic information about percentage changes in breach time and outflow, the actual mechanisms of erosion development in cracks and the role of cracks in accelerating the headcut and breach development process are not given much focus. Overall, the paper is written in very fractured English and is difficult to read and understand in many passages.

Response:

1. Hanson et al. (2005) conducted large-scale overtopping tests to understand the rate, timing, and processes of dam failures. Their results on the overtopping tests illustrate how certain soil properties influence the timing and rates of the observed erosion processes. The increase of the erosion rates (vertically, longitudinally and horizontally) in the three orders of magnitude, show a direct correlation to the compaction water content and soil texture. They mainly worked for correlating the $K_d$ (erodibility factor) with the headcut migration rate and breach widening models. Although compaction water content and energy along with type of soil strongly correlates with the erosion processes, they neglected this effect from further analysis in their correlation process while formulating the mathematical models. Also, the results include more than one type of soil, whereas the hydraulic loading between the seven tests is varied due to the different embankment configurations. As we know, with change in location of a dam, the properties of soil changes. Thus any study on embankment failures remains inconclusive even after so many experiments. The soil used in the present work is taken

directly from the dam construction site. The experiments performed in this research, tested the behavior of soil with respect to compaction moisture content and energy that adds more knowledge to the literature. A new dimension to the dam failure study is tried to be introduced in this paper.

2. In the revised manuscript the data is presented in the way so as to facilitate the comparisons as suggested by the reviewer. More lab tests on unconfined compression of soil are performed and added in the revised manuscript for comparison.

3. On reviewer suggestion, we try to run the WinDAM software using experimental data to facilitate the comparison between software results and the experimental results. We have put enormous efforts into running the software WinDAM in the present work, but finally dropped the idea of comparing the software results with the experimental outputs due to some unavoidable reasons. We have also tried with Mike 11 (erosion based) for our analysis but the handicap of this software is incapable of handling the effect of moisture content and the dam dry density.

4. The experiments performed in Phase 2 are basically trial runs to investigate whether the cracks developed in embankments affect the breach parameters or not. We wish to take of more rigorous study on the aspect of the influence of cracks on actual mechanisms of erosion in the next spiral of our experiments.

5. We agree with your comment on fractured English in many passages. We have undertaken thorough editing of the paper and improved the quality of English substantially.

Specific comments:

Embankment heights are reported inconsistently throughout the paper. Values of 0.3 m, 0.5 m, and 0.6 m appear in various places. Some tests are characterized as small-scale and others as large scale, despite apparently small differences in embankment height. Response: The height of embankment is kept constant in all the experiments

which are 0.3 m. Phase 1 and Phase 2 experiments were performed in two different flumes. The value 0.5 m is the flume depth in Phase 2 experiments, while the value 0.6 m is flume depth in Phase 1 experiments.

Technical corrections:

The works of Hanson should be more fully described. They were lab tests of constructed embankments, not studies of real dam failures. The text gives so little information that a reader could easily infer the latter. Page 4, lines 20-23 talk of two "crucial scenarios", one in which stored reservoir water is not contributing to outflow, a second in which stored water is released through the breach. However, what follows does not describe two scenarios, but three phases that seem to apply to all of the tests. The two "crucial scenarios" seem to never be mentioned again, suggesting that they were not so crucial. This is disturbing for the reader who feels they have missed an important point. Units for the dimensions of the tamper equipment are not given.

Response to technical comments:

1. The works of Hanson is described to the required extent in this paper. Hanson mainly works for correlating the Kd (erodibility factor) with the headcut migration rate and breach widening models. 2. The word "crucial-scenarios" is omitted in the present revision. 3. The dimensions of tamper equipment is now added to the text under the section 5.2: Experimental Study in Phase 1 (Page 5, line no. 7)

---

## Referee Comment (RC2) · Anonymous Referee #2 · 10 Jun 2018

**General comments**

The present work studies the effect of construction parameters including compaction effort, compaction moisture content, and the moisture content at the time of failure on the embankment breaching due to overtopping. The contribution of this study is useful and interesting. My studies in this field shows that the some of the parameters such as effect of compaction have been studied in the previous researches (e.g. Ali Asghari Tabrizi, Ezzat Elalfy, Mohamed Elkholy, M. Hanif Chaudhry & Jasim Imran (2016): Effects of compaction on embankment breach due to overtopping, Journal of Hydraulic Research, DOI: 10.1080/00221686.2016.1238014.). But the effect of crack in the hydraulics of embankment breaching is novel. The reported data have good agreements with physics of this phenomenon. According to my viewpoint, the measured data were not reported in scale of a research paper. For example, the embankment surface profiles at various times were not shown in the results. The main strong section of the present study which is about the effect of cracks in the breaching process was not discussed well. Also, to ensure the reliability of the measurements, the repeatability of the tests must be checked by considering the various parameters for all cases prior to analysis the results. In this work, the test repeatability was not discussed.

**Specific comments**

The present study is about the embankment breaching which is made by the cohesive soils. However, the effect of the cohesion on the hydraulic of breaching was not studied.

**Technical corrections**

The literature review of present paper could be rewritten. Some major studies must be reviewed, and the novelty of present is expressed clearly. The temporal embankment profiles for each case were not presented, while it is very important for such studies. Finally, a precise indication must be provided for the impact of the mentioned parameters on the embankment breaching.

---

## Author Comment (AC2) · 19 Jul 2018

Sachin Dhiman and Kanhu Charan Patra

sach.dhiman1988@gmail.com

General comments The present work studies the effect of construction parameters including compaction effort, compaction moisture content, and the moisture content at the time of failure on the embankment breaching due to overtopping. The contribution of this study is useful and interesting. My studies in this field shows that the some of the parameters such as effect of compaction have been studied in the previous researches (e.g. Ali Asghari Tabrizi, Ezzat Elalfy, Mohamed Elkholy, M. Hanif Chaudhry & Jasim Imran (2016): Effects of compaction on embankment breach due to overtopping, Journal of Hydraulic Research, DOI: 10.1080/00221686.2016.1238014.). But the

effect of crack in the hydraulics of embankment breaching is novel. The reported data have good agreements with physics of this phenomenon. According to my viewpoint, the measured data were not reported in scale of a research paper. For example, the embankment surface profiles at various times were not shown in the results. The main strong section of the present study which is about the effect of cracks in the breaching process was not discussed well. Also, to ensure the reliability of the measurements, the repeatability of the tests must be checked by considering the various parameters for all cases prior to analysis the results. In this work, the test repeatability was not discussed.

Response We are thankful to Reviewer for the valuable comments. As suggested, the embankment surface profiles at various times are added for Phase 1 experiments (P1.2, P1.3, and P1.4). The experiments performed in Phase 2 are new to this field and are basically trial runs to investigate whether the cracks developed in embankments affect the breach parameters. The trial runs itself present sufficient difference in final outputs and we wish to take up more rigorous study on the aspect of the influence of cracks on actual mechanisms of erosion and the repeatability in the next spiral of our experiments.

Specific comments The present study is about the embankment breaching which is made by the cohesive soils. However, the effect of the cohesion on the hydraulic of breaching was not studied.

Response All the embankments are constructed for same lean clay - CL type soil (cohesive in nature). Our main focus was to vary the compaction moisture content and compaction energy and their influence on the breach parameters. A new factor is introduced in the revised manuscript, which is dependent on the relative dry density and relative moisture content of the embankment. The effect of cohesion of soil on breaching was not focused in the current work.

Technical corrections The literature review of the present paper could be rewritten.

Some major studies must be reviewed, and the novelty of present is expressed clearly. The temporal embankment profiles for each case were not presented, while it is very important for such studies. Finally, a precise indication must be provided for the impact of the mentioned parameters on the embankment breaching.

Response The literature review is improved and focused by considering more studies on this field including the work of Tabrizi et al. (2016) in the revised manuscript. Temporal embankment profiles for Phase-1 experiments are now presented. The conclusions are rewritten with more clarity describing the impact of compaction moisture content, compaction energy and embankment cracks on breach parameters in the revised manuscript.

Please also note the supplement to this comment:
https://www.nat-hazards-earth-syst-sci-discuss.net/nhess-2017-383/nhess-2017-383-AC2-supplement.pdf